# ANOMALYLMM: BRIDGING GENERATIVE KNOWLEDGE AND DISCRIMINATIVE RETRIEVAL FOR TEXT-BASED PERSON ANOMALY SEARCH

## ABSTRACT

Text-based person anomaly search requires fine-grained alignment between language queries and subtle visual cues, a challenge amplified by the long-tailed nature of anomalous behaviors. While Large Multimodal Models (LMMs) offer powerful visual understanding, their generative pre-training is fundamentally misaligned with the discriminative needs of this retrieval task. Adapting them without costly fine-tuning remains a significant hurdle. We introduce **AnomalyLMM**, a training-free, coarse-to-fine framework that effectively repurposes generative LMMs for zero-shot anomaly retrieval. Our method first uses a general retrieval model to produce an initial ranking. To refine this list, we introduce a novel cloze-based re-ranking mechanism with three steps. The first step **Cloze Generation** converts the text query into a "fill-in-the-blank" prompt. Next, **Cloze Completion** compels the LMM to focus on specific visual regions and generate a description of the potential anomaly. The final step **Comparison & Re-ranking** calculates semantic alignment between the LMM's generated completion and the original query, which serves as a powerful re-ranking score. Experiments on the PAB dataset show that AnomalyLMM surpasses the competitive baseline by +0.96% Recall@1 accuracy. Crucially, our method provides highly interpretable visual-textual alignments without any task-specific training, a vital feature for real-world deployment. The code will be made publicly available.

## 1 INTRODUCTION

Ensuring public safety is a growing priority in smart surveillance systems, driving the need for efficient text-based person anomaly search, a task that retrieves individuals exhibiting abnormal behaviors using natural language queries. Unlike generic text-based person search (Zheng et al., 2020; Yang et al., 2023), this task focuses on identifying subtle, often rare anomalous actions (*e.g.*, falling, collision, or being hit) described in text. Such capability is critical for real-world applications, from crime prevention to emergency response. However, it remains underexplored due to the lack of dedicated benchmarks and the inherent complexity of aligning textual descriptions with fine-grained anomalous behaviors in videos.

This task introduces two key challenges: **(1) Fine-grained cross-modal alignment:** Anomalies are often described with nuanced semantics (*e.g.*,"staggering while clutching their chest"), requiring precise localization of transient visual cues. Existing methods (Yang et al., 2025; Zeng et al., 2024) struggle with such alignment, as they primarily focus on coarse-grained attributes (*e.g.*, clothing or pose). **(2) Sparse anomaly samples:** Real-world anomaly data are long-tailed and diverse, making it challenging to train or fine-tune models effectively. While recent Large Multi-modal Models (LMMs) (Li et al., 2023; Alayrac et al., 2022; Achiam et al., 2023; Yang et al., 2024) show strong general-purpose visual-language reasoning, they are primarily designed for generative tasks rather than discriminative retrieval. Consequently, directly employing LMMs in anomaly search has two deficiencies: (1) There exists a notable domain gap between the general pretraining objectives and the specific demands of anomaly retrieval. (2) There remains a lack of efficient adaptation strategies to effectively leverage LMMs for deployment in few-shot or resource-constrained scenarios, limiting their applicability to anomaly search (see Figure 1 (*left*)).

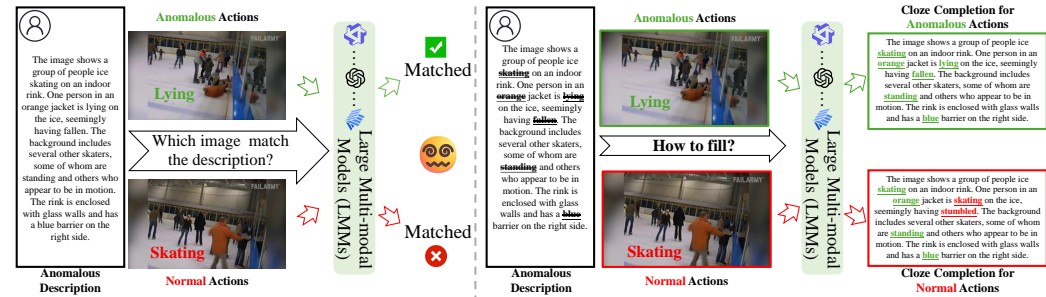

Figure 1: *Motivation of AnomalyLMM.* (*left*) Directly matching candidate images with anomalous actions is challenging for LMMs. This difficulty is twofold: visually, the cues are often subtle and local; textually, the critical descriptors usually constitute only a small semantic fraction of a lengthy, complex query. (*right*) Cloze completion is an effective adaptation strategy for LMMs, allowing LMMs to focus on relevant objects and establish fine-grained cross-modal alignment between text and images. Green completions in the upper-right box indicate that completions from LMM align with the anomalous description, while red completions denote mismatches.

In this work, we propose AnomalyLMM, the first framework that strategically adapts LMMs for text-based person anomaly retrieval. Instead of directly relying on LMMs to retrieve images based on the given text query, we propose a coarse-to-fine retrieval pipeline that first narrows the candidate space and then progressively refines the search using LMMs' reasoning abilities (see Figure 1 (Right)). Specifically, we first employ a general text-to-image retrieval model to produce an initial list of candidate images given a natural language anomaly query. These general retrieval models, trained for global cross-modal alignment, are efficient at coarse-level retrieval and serve as a strong foundation for subsequent reasoning. **To move beyond coarse retrieval and enable more fine-grained alignment, we reformulate anomaly search as a masked cloze-style reasoning task.** Key verbs and color attributes in the original query are replaced with structured placeholders (< VERB >, < COLOR >) using a Large Language Model (LLM). This yields a masked query that disentangles subtle behavioral semantics and visual appearance. Each candidate image is paired with a masked query and fed into an LMM for cloze completion, thereby prompting the model to focus on subtle behavioral, visual appearance, and other anomaly-related cues. This cloze-style completion transforms each candidate image into a query-specific textual description. The intuition is that a correctly paired image inherently contains the visual cues (*e.g.*, action verbs or appearance attributes) required to fill the masked placeholders, whereas irrelevant candidates lack such cues and therefore are more likely to produce incorrect completions or generic tokens such as 'UNKNOWN'. To avoid hallucinations, we constrain the LMM to output UNKNOWN when uncertain, ensuring that completions are grounded in the image. Next, the original query is compared with all image-specific completions using another Large Language Model (LLM). This comparison is performed semantically, considering language variability. Verbs with similar meanings (*e.g.*, "balancing" and "stumbling") or colors with close hues (*e.g.*, "dark" and "gray") are treated as matched. Based on the number of matched placeholders, we derive a semantic re-ranking list of candidate images. Finally, we jointly take this re-ranking list and the initial ranking list into consideration, yielding the final ranking list. Together, these steps form a unified pipeline that reformulates the anomaly retrieval task as a masked completion and re-ranking process. AnomalyLMM enables LMMs to operate in a reasoning-centric fashion without requiring additional training or fine-tuning. Our primary contributions are as follows:

- **A training-free, coarse-to-fine framework, AnomalyLMM, that adapts LMMs for anomaly search.** In particular, we successfully repurpose the generative knowledge of pre-trained LMMs for a discriminative retrieval task without any fine-tuning. Our framework achieves this through a synergistic three-step pipeline: **Cloze Generation**, **Cloze Completion**, and **Comparison & Re-ranking**, which associates the discriminative textual descriptors with the local visual cues.

- **State-of-the-art performance and verifiable interpretability on the PAB benchmark.** Our method not only sets a new state-of-the-art by outperforming existing methods by 0.96% Recall@1 accuracy, but also provides crucial, interpretable alignments between textual queries and visual behavior, validating its effectiveness and potential for practical deployment.

## 2 RELATED WORK

**Anomaly Detection in Public Safety.** Traditional anomaly detection approaches rely on hand-crafted features (Wang & Miao, 2010) or reconstruction-based paradigms (Hasan et al., 2016), identifying anomalies as statistical deviations from learned normal patterns. Recent advances adopt deep learning for spatio-temporal modeling, including graph networks (Scarselli et al., 2008) for crowd dynamics and self-supervised learning (Gui et al., 2024) to exploit unlabeled behavioral data. While methods (Sultani et al., 2018) achieve strong video-level anomaly classification, they operate in an unsupervised manner, lacking explicit mechanisms to align visual anomalies with textual queries. Following the general person re-identification (Zheng et al., 2017) and text-guided person search (Zheng et al., 2020; Bai et al., 2023), text-anchored approaches (Shen et al., 2023; Wang et al., 2025) attempt cross-modal retrieval but face two limitations: (1) coarse-grained semantics, such as attributes (clothing or pose) or the binary tags (normal/abnormal). Such annotation limitation overlooks subtle behavioral cues (*e.g.*, *staggering* vs. *falling*), and (2) bias toward normal behaviors due to long-tail distributions. The system learns more normal behaviors, and is prone to predict normal actions. To address the data scarcity, Yang *et al.* introduce the PAB dataset that pioneers text-based person anomaly search to cover more anomaly action types via natural language description (Yang et al., 2025), yet state-of-the-art solutions (Zeng et al., 2024; Yang et al., 2025) rely on supervised fine-tuning, which struggles with rare or unseen anomalies. Therefore, it remains challenging to design a scalable system for addressing unseen anomaly actions and giving quick responses, which is key to smart city management. In an attempt to generalize anomaly detection systems, our work addresses these gaps by unifying generative knowledge from LMMs with discriminative retrieval.

**Fine-grained Cross-modal Retrieval.** This task has gained increasing attention due to its ability to align and retrieve highly specific content across different modalities (*e.g.*, images and text). Early works in cross-modal retrieval primarily focused on coarse-grained alignment, learning shared representations using methods like canonical correlation analysis (Andrew et al., 2013) and deep metric learning (Lu et al., 2017). However, these approaches often struggle with fine-grained distinctions due to their limited ability to capture subtle inter-modal correspondences. Recent advances leverage fine-grained feature interactions to improve retrieval precision. For instance, Faghri *et al.* introduces a hierarchical alignment framework that matches local regions in images with phrases in text (Faghri et al., 2018). Ji *et al.* employs cross-modal attention to dynamically focus on discriminative sub-elements (Ji et al., 2021). Transformer-based models (Gorti et al., 2022; Qu et al., 2024) further enhance fine-grained retrieval by modeling long-range dependencies across modalities. Another line of research explores contrastive learning for fine-grained cross-modal alignment. Methods like (Radford et al., 2021) employ instance- and patch-level contrastive losses to enforce both global and local consistency. Additionally, Li *et al.* propose to leverage test-time adaptation to narrow the modality gap between the query and the gallery. Despite progress, challenges remain in handling highly nuanced queries and mitigating modality-specific biases. Our work addresses these limitations by introducing a training-free cross-modal re-ranking mechanism that enhances fine-grained discriminability while maintaining computational efficiency.

**Large Multi-modal Models for Discriminative Tasks.** Recent LMMs (Chen et al., 2024b; Yao et al., 2024; Chen et al., 2022) unify vision-language understanding through large-scale pre-training on aligned image-text pairs. Some works also explore audio (Deshmukh et al., 2023), video (Maaz et al., 2023; Chen et al., 2024a) and multiple data formats (Wu et al., 2024). The typical models like CLIP (Radford et al., 2021), BLIP (Li et al., 2023) based on contrastive learning enable zero-shot cross-modal retrieval, while generative LMMs (*e.g.*, GPT-4V (Hurst et al., 2024), LLaVA (Liu et al., 2023)) excel in open-world reasoning via instruction tuning. However, adapting these models for fine-grained anomaly search poses unique challenges as follows. (1) *Semantic Granularity Gap.* Generative LMMs prioritize holistic scene understanding over localized behavioral anomalies (Radford et al., 2021); (2) *Adaptation Inefficiency.* Existing strategies (Zhang et al., 2023; Rong et al., 2025) require full fine-tuning or adding additional parameters for tuning, such as LoRA (Hu et al., 2022) and WoRA (Sun et al., 2025), which are still prohibitively expensive for sparse anomaly data and large GPU memory consumption. While prompt-based methods (Zhou et al., 2022) improve attribute alignment, they fail to capture dynamic anomaly saliency (*e.g.*, temporal *collision* cues). Our framework innovates by (i) repurposing LMMs' world knowledge for anomaly localization without parameter updates, and (ii) introducing a training-free adaptation protocol to bridge generative and discriminative paradigms.

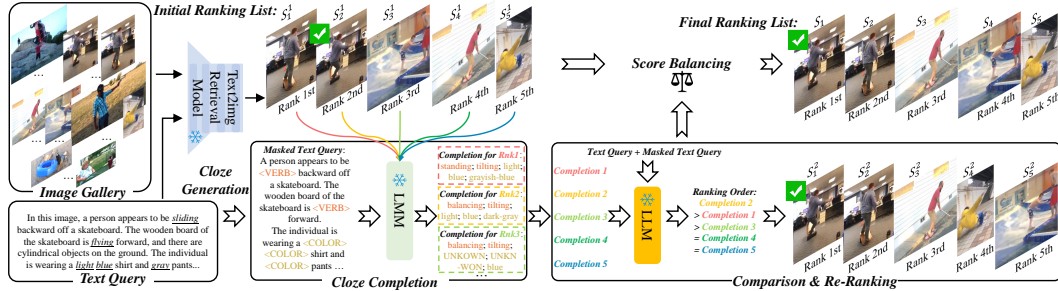

Figure 2: *Overall pipeline of the proposed AnomalyLMM*. First, given a text query about the target person with anomaly action, we apply an off-the-shelf text2image retrieval model to obtain the initial ranking list (*left*). Next, the verbs and colors in the text query are replaced with placeholder tokens $<$ VERB $>$ and $<$ COLOR $>$ via an LLM, generating clozes (*Cloze Generation*). According to the candidate images, a pre-trained LMM is then prompted to complete these clozes by predicting masked tokens for each image in the initial ranking list (*Cloze Completion*). LMMs generate semantically similar completions when provided with the correctly paired image, but produce dissimilar ones for all other candidates. Subsequently, the completions for each image are compared and re-ranked by an LLM in a semantic equivalent manner, producing the re-ranking list (*Comparison & Re-Ranking*). Finally, the initial and re-ranking lists are combined through score balancing to produce the final ranking list (*upper-right*).

## 3 METHODOLOGY

Our method is a coarse-to-fine retrieval pipeline. At the coarse stage, we first leverage an off-the-shelf text-to-image retrieval model to produce an initial ranking of top-$N$ candidate frames. The fine stage consists of three key steps: Cloze Generation, Cloze Completion, and Comparison & Re-ranking. The final ranking result is obtained by fusing the initial ranking list (from the coarse stage) with the re-ranking result (from the fine stage). A brief overview of the pipeline is shown in Figure 2. We are inspired by the cloze task to bridge the gap between the large models pretrained on generative tasks and the discriminative retrieval demands.

### 3.1 INITIAL RANKING GENERATION

The coarse-to-fine strategy starts with an initial filtering step. Specifically, we employ a dedicated text-to-image retrieval model, *i.e.*, X$^2$VLM (Zeng et al., 2024). This model is trained to match visual content with natural language descriptions and has been widely used in general retrieval tasks due to its efficiency and discriminative power. Given a natural language anomaly query, we retrieve the top-$N$ images from the dataset. These candidates form an initial ranking list which is expected to include relevant results and exclude most irrelevant ones. By reducing the input space for subsequent processing, we not only improve computational efficiency but also allow subsequent LMMs to operate in a more focused, context-rich setting. While LMMs excel at contextual reasoning and text-image generation, their native formulation is not directly optimized for discriminative retrieval tasks. Thus, instead of directly aligning the text query with visual content using LMMs, we first convert the images in the initial ranking list into text descriptions via cloze generation and completion. This allows us to perform the comparison entirely in the textual domain by matching the generated descriptions with the original query.

### 3.2 CLOZE GENERATION

We leverage LLMs to generate two distinct types of masks from the input text query: verb masks and color masks. These masks are designed to disentangle action semantics and appearance attributes, enabling the model to focus more effectively on salient cues relevant to anomaly retrieval. For the verb mask, we utilize an LLM (e.g., *QVQ-Max*) to identify all action-related or lexical verbs within the query. Each detected verb is then replaced with a standardized placeholder token, `< VERB >`. For the color mask, we prompt the same LLM to extract all color-related descriptions that refer to either the person or their surrounding environment. Identified color terms are similarly substituted with `< COLOR >` tokens one by one. The specific prompts used for action-related verbs are illustrated in Figure 3 (a) (The prompt for color-related descriptions is also similar). By combining these

**(a) Cloze Generation Prompt**

You are a linguistic analyzer. Your task is to extract all verbs and verb phrases from the input sentence.

**Steps**:
1. Identify all meaningful verbs and verb phrases for actions, ignoring auxiliary/modal verbs.
2. Exclude any non-verb words from verb phrases, including nouns, adjectives, adverbs, prepositions, and pronouns.
3. Phrasal verbs: Only include the verb in a phrasal verb, and exclude any prepositions or nouns.
4. Exclude any words or phrases that do not describe an action.

Examples:
…

**(b) Cloze Completion Prompt**

Based on the VISUAL EVIDENCE in the image, replace each <VERB> and <COLOR> slot in the given template with exactly one contextually appropriate word.

**Think following these steps**:
1. Analyze the subjects' body positions, spatial relationships, and environmental context in the input image.
2. Identify only the observable actions or colors supported strictly by visual cues (\eg limb angles, movement, color visible in the image).
3. Do NOT guess or infer any information from the input text or external knowledge.
4. If no visual evidence supports a replacement, or if the described objects do not appear in the image, use the word 'UNKNOWN' for that slot.

**Examples**:
…

**(c) Comparison & Re-ranking Prompt**

**Background**:
There are three types of inputs:
1. A template sentence that contains <COLOR> and/or <VERB> placeholders.
2. A description that fills those placeholders.
3. N options (labelled A, B, C …). Each option is a sentence that also fills the placeholders.
Treat all inputs as visual descriptions of images.

**Task**:
Rank the options by how well their replacement words for each placeholder match those in the description. Count both exact and visually / functionally equivalent matches.

**Equivalence rules**:
<COLOR>:
1. Same hue or obvious tint/shade is equivalent.
2. Near-hue neighbors count as matches.
3. Distinct hues are not equivalent.
<VERB>:
1. Evaluate the visible action, not the literal wording.
2. Two verbs are equivalent when they depict the same physical motion and the same state/instability.
3. Do not favor exact lexical matches over visually equivalent verbs.

**Examples**:
…

Figure 3: *Prompt for different steps in AnomalyLMM.* (a) Cloze generation prompt. This prompt is designed for the LLM to extract verbs that describe actions from input sentences. A similar prompt is used for identifying color-related descriptions. (b) Cloze completion prompt. This prompt guides the LMM to use visual evidence from images to fill the <VERB> and <COLOR> tokens in the masked text query with contextually appropriate words or 'UNKNOWN' when no visual support exists. (c) Comparison & Re-ranking prompt. This prompt directs the LLM to compare and rank the generated completions based on semantic equivalent rules, considering both exact and visually / functionally similar matches.

two types of masks, we obtain a masked text query, which serves as part of the input to the LMM in the subsequent anomaly-aware retrieval process.

## 3.3 CLOZE COMPLETION

Given a masked text query containing structured placeholders, *i.e.*, $< VERB >$ and $< COLOR >$, we employ an LMM to perform cloze-style completion for each image in the initially retrieved ranking list. Specifically, for each candidate image in the ranking list, the LMM is prompted with the image and the obtained masked query as context, and tasked with filling in the missing placeholders based on visual evidence. This process is repeated for all $N$ images in the initial ranking list, resulting in $N$ distinct cloze completions tailored to the visual content of each image. To avoid hallucinated completions, we explicitly instruct the LMM to output ``UNKNOWN'' if it cannot confidently infer the placeholder content from the given image. Meanwhile, this constraint also helps alleviate the tendency to generate context-independent completions solely based on textual priors rather than grounding them in visual evidence. An example prompt used for this cloze completion step is illustrated in Figure 3 (b). These completions are then used for fine-grained re-ranking in the next step. Here, our primary objective is to achieve **synonyms** completions rather than exact word-for-word completions. Given a correctly paired image, an LMM can complete the cloze task with high semantic fidelity by generating the exact word or its synonyms. In contrast, when the image is unpaired, the model generally fails to produce any semantically appropriate alternatives.

## 3.4 COMPARISON & RE-RANKING

**Comparison.** Each image in the initial ranking list is associated with a cloze completion generated from the masked query. To assess the relevance of each image, we compare its completion with the original text query using an LLM-based semantic comparison. Specifically, the LLM receives three inputs: the original query, the masked query (with placeholders), and the set of generated

completions. By including the masked query, we guide the LLM to focus on the replaced tokens, enabling a more targeted comparison at the locations of interest. Given the inherent ambiguity and variability in natural language, we perform semantic rather than literal comparison. That is, instead of checking for exact word matches, we evaluate whether the generated completions convey concepts consistent with those in the original query. For verbs, actions that are semantically similar (*e.g.*, "balancing" vs. "sliding") are treated as matched when they describe equivalent behaviors, such as a person on a skateboard poised to slide. Similarly, color terms like "gray" and "dark" are treated as equivalent when referring to the same clothing item. This semantic alignment helps ensure that images are ranked not only by surface-level similarity but by conceptual consistency with the original text query.

**Re-Ranking.** Following the semantic comparison, we prompt the LLM to re-rank all completions in descending order based on the number of semantically matched placeholders. To account for cases where multiple completions exhibit the same level of semantic consistency with the original query, we also instruct the LLM to identify equivalence relationships during re-ranking. In such cases, completions with equal semantic alignment are assigned the same rank. The full prompt used for semantic comparison and re-ranking is provided in Figure 3 (c). To obtain the final ranking result, we combine the rankings from the initial ranking list and the re-ranking result. The initial ranking is based on the retrieval scores predicted by the text-to-image model, whereas the re-ranking result is based purely on semantic alignment and provides only an ordered list without associated scores. To enable fusion across the two lists, we assign scores to the re-ranking result using an exponential decay function with base $\beta$. Specifically, for a completion ranked at position $n$, its score is computed as: $S_n^2 = \beta^n$. The final ranking score is then computed as a weighted combination of the two lists:

$$S_n = \alpha_1 S_n^1 + \alpha_2 S_n^2, \quad n \in [0, N-1], \tag{1}$$

where $\alpha_1$ and $\alpha_2$ are balancing weights, $S_n^1$ and $S_n^2$ represent the scores from the initial and re-ranking lists, respectively, and $N$ is the number of retrieved candidates in the initial list.

**Discussion-1. What are the advantages of cloze generation and completion?** Inspired by recent successes in general text-based person retrieval (Niu et al., 2025; Tan et al., 2024; He et al., 2024) and target region localization in LMMs (Zhang et al., 2025; Liu et al., 2024), we propose cloze generation and completion to better exploit the reasoning capabilities of LMMs and guide their attention toward anomalous regions in the image. Compared with previous methods, our cloze generation and completion offer two key advantages: (i) Both modules are *training-free* and do not require any additional training or fine-tuning, making them efficient and effective for text-based person anomaly search, especially since real-world anomaly data is scarce and insufficient for fully training LMMs; (ii) Cloze generation and completion actively prompt LMMs to infer missing information based on both the image content and the surrounding textual context. This naturally encourages the model to attend to visually relevant regions aligned with the anomaly description, enhancing interpretability and retrieval precision.

**Discussion-2. What are the advantages of comparison & re-ranking?** The proposed comparison & re-ranking method has three advantages. (1) Precise alignment between text query and visual content is critical for anomaly search. Instead of fusing cross-modal features, our method compares cloze completions in the textual domain, enabling fine-grained, interpretable alignment that fully leverages the language understanding capabilities of LLMs. (2) Due to the inherent ambiguity in natural language, identical actions can be described using different words. Instead of surface-level or lexical matching, our module performs comparison in a semantically equivalent manner, making it more robust to diverse phrasings of the same concept. (3) Unlike methods that predict absolute similarity scores (Zeng et al., 2024; 2022), we focus on comparing and directly predicting the relative ranking. By restricting comparison to masked positions (*e.g.*, verbs, colors) and jointly considering all candidates in the initial list, the LLM benefits from global context to make more consistent and context-aware reordering decisions.

## 4 EXPERIMENT

**Implementation Details.** We adopt the official CMP (Yang et al., 2025) and the widely-used $X^2$VLM (Zeng et al., 2024) as two baseline models trained on the PAB dataset. The models are all fine-tuned with a batch size of 22 using AdamW (Loshchilov & Hutter, 2019) and a weight decay of $1 \times 10^{-2}$. The starting learning rate is set to $5 \times 10^{-5}$ and scheduled by a StepLR policy, with

Table 1: Comparison of proposed method with competitive methods on the PAB (Yang et al., 2025). Integrating representative text-to-image retrieval models into the proposed method consistently improves performance. Both CMP-baseline and $X^2$VLM benefit from notable gains in Recall@1 (R@1) and mAP when combined with our approach.

| Method | R@1 | mAP |
|---|---|---|
| APTM (Yang et al., 2023) | 72.14 | 82.78 |
| IRRA (Jiang & Ye, 2023) | 76.39 | 86.33 |
| CLIP (Radford et al., 2021) | 77.60 | 87.35 |
| RaSa (Bai et al., 2023) | 80.79 | 89.20 |
| X-VLM (Zeng et al., 2022) | 81.95 | 89.86 |
| CMP (Yang et al., 2025) | 83.06 | 90.41 |
| CMP-baseline (Yang et al., 2025) | 69.51 | 81.06 |
| CMP-baseline + *Ours* | 73.15 (+3.64) | 83.64 (+2.58) |
| $X^2$VLM (Zeng et al., 2024) | 83.77 | 90.43 |
| $X^2$VLM + *Ours* | **84.73 (+0.96)** | **90.89 (+0.46)** |

a warmup of $5 \times 10^3$ iterations and decays applied at $2 \times 10^4$ and $3.5 \times 10^4$ iterations. We employ QWQ-Max for the cloze generation and the comparison and re-ranking steps, and use QVQ-Max for cloze completion. **In this work, we do not pursue the best text-to-image model for coarse retrieval, but focus on further improvement by leveraging large models to narrow down the final candidate ranges.**

**Dataset.** PAB (Yang et al., 2025) is the first large-scale benchmark for the text-based pedestrian anomaly search, covering diverse scenarios paired with their anomalous counterpartsfor the same identity. The test set contains 1,978 real-world image-text pairs with manually verified anomalies, providing a rigorous evaluation platform. We adopt standard retrieval metrics. Recall@K computes the fraction of queries where the true match appears in the top-K results, reflecting real-world usability. We mainly report the Recall@1 (R@1) considering the fine-grained retrieval demands. Mean Average Precision (mAP) measures overall ranking quality by averaging precision across recall thresholds, emphasizing correct top-rank retrievals.

### 4.1 COMPARISON WITH COMPETITIVE METHODS

**Quantitative Results.** We evaluate the two representative text-to-image retrieval models, *i.e.*, CMP (Yang et al., 2025) and $X^2$VLM (Zeng et al., 2024) in AnomalyLMM and compare against other competitive approaches (see Table 1). All competitive methods are fine-tuned on the PAB dataset (Yang et al., 2025). The proposed method achieves notable improvements over strong baselines (Yang et al., 2025; Zeng et al., 2024). Specifically, we observe $+3.64\%$ Recall@1 accuracy, $+2.58\%$ mAP on CMP baseline, and $+0.96\%$ Recall@1 accuracy, $+0.46\%$ mAP on $X^2$VLM baseline. These results validate the effectiveness of cloze generation, completion, and comparison & re-ranking in the proposed method and facilitate the text-based person anomaly search.

**Qualitative Results.** We present four qualitative examples of text-based person anomaly search (see Figure 4). For the first three cases (a-c), our method effectively retrieves the corresponding gallery images based on the query texts. In Figure 4 (a), actions such as "waveing" and "fall" are critical. The cloze completion for the top-ranked gallery replaces "waveing" with "swinging," which is semantically similar, while other completions like "adjusting" are reasonable but less semantically aligned. However, for the action "fall", our method fails to predict accurate completions across all retrieved galleries because the correctly paired image only shows the tendency of an imminent fall. This process is visually similar to a leap. Similar patterns of partial semantic matches and occasional failures appear in Figure 4 (b) and Figure 4 (c). In Figure 4 (d), our method does not retrieve the correct gallery image despite semantically matching completions in the top results. This is mainly due to the ambiguity in the query text, which presents a challenging scenario even for human annotators to identify the ground truth.

### 4.2 ABLATION STUDIES AND FURTHER DISCUSSION

**Effect of primary components.** We conduct ablation studies with $X^2$VLM as the baseline and study our primary components (see Table 2 (a)). We treat cloze generation and completion as a single module. For the effect of cloze generation and completion ($w$ cloze completion in Table 2 (a)), we add this module into the baseline and re-rank generated completions based on the cosine similar-

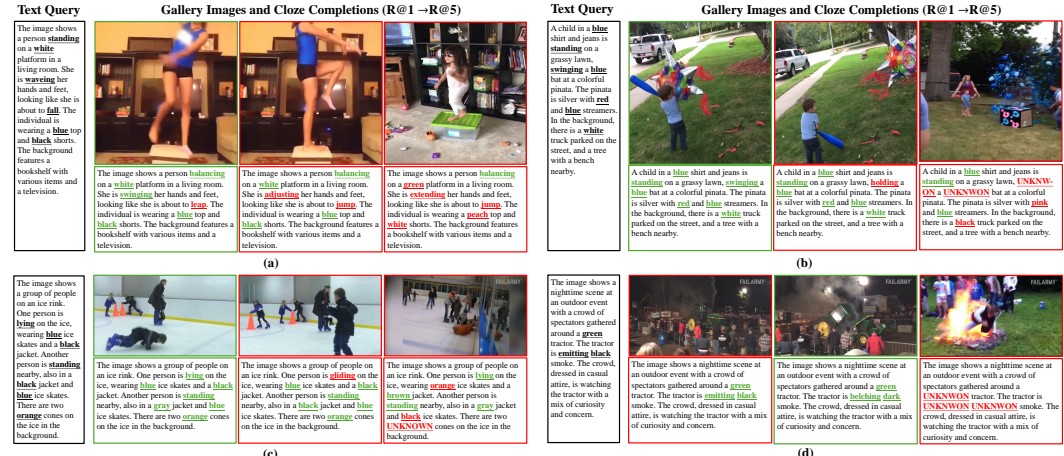

Figure 4: Qualitative results on the PAB dataset (Yang et al., 2025). (a)-(c): successful cases; (d): a failure case. Given text queries (left), matched gallery images and cloze completions are in green, and mismatched are in red (right). In (d), the presence of highly similar objects exhibiting comparable appearances and actions consistent with the query text results in a reasonable failure. Zoom in for a better visualization.

Table 2: Ablation studies. (a) Ablation study of primary components. Applying both yields the best performance. (b) Ablation study of weighting strategies for re-ranking. Exponential decay with base $0.5$ achieves the best performance. (c) Ablation study of LLM or LMM for different steps. (d) Ablation study on the number of initial candidates. A length of three candidates achieves the best performance. (e, f) Ablation study of hyper-parameters ($\alpha_1$, $\alpha_2$) for score balancing.

(a)

| Method | Cloze Gen. & Completion | Comparison & Re-Ranking | R@1 | mAP |
|---|---|---|---|---|
| Baseline | ✗ | ✗ | 83.77 | 90.43 |
| $w$ Cloze Completion | ✓ | ✗ | 84.12 | 90.60 |
| $w$ Re-Ranking | ✗ | ✓ | 84.22 | 90.65 |
| Ours | ✓ | ✓ | **84.73** | **90.89** |

(b)

| Strategy | Hyper-parameter | Performance | |
|---|---|---|---|
| | | R@1 | mAP |
| Exponential $\beta$ | 0.70 | 84.63 | 90.86 |
| | 0.50 | **84.73** | **90.89** |
| | 0.30 | 84.73 | 90.88 |
| | 0.20 | 84.68 | 90.88 |
| Linear $d$ | -0.30 | 84.68 | 90.88 |
| | -0.20 | 84.73 | 90.88 |
| | -0.10 | 84.68 | 90.88 |

(c)

| LLM/LMM Selection | | | Performance | |
|---|---|---|---|---|
| Cloze Gen. | Cloze Comp. | Re-Ranking | R@1 | mAP |
| QWQ-Max | QVQ-Max | Qwen2.5-3B | 83.67 | 90.37 |
| QWQ-Max | QVQ-Max | Qwen2.5-7B | 83.87 | 90.48 |
| QWQ-Max | Qwen2.5-VL-7B | QWQ-Max | 84.13 | 90.62 |
| QWQ-Max | Qwen2.5-VL-3B | QWQ-Max | 84.22 | 90.65 |
| Qwen3-8B | QVQ-Max | QWQ-Max | 84.52 | 90.80 |
| QWQ-Max | QVQ-Max | Qwen3-8B | 84.63 | 90.86 |
| QWQ-Max | QVQ-Max | QWQ-Max | **84.73** | **90.89** |

(d)

| # Initial Candidate | R@1 | mAP |
|---|---|---|
| 2 | 84.63 | 90.85 |
| 3 | **84.73** | **90.89** |
| 4 | 84.68 | 90.88 |
| 5 | 84.27 | 90.66 |

(e)                                             (f)

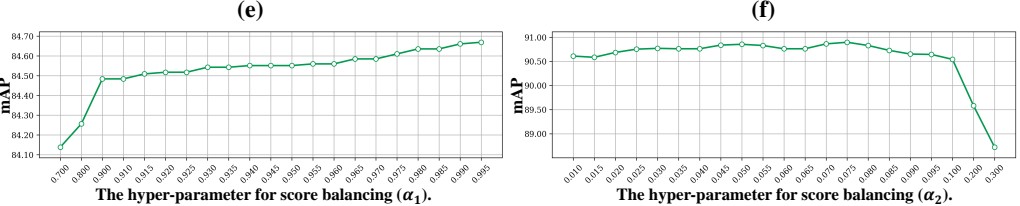

ity between `all-MiniLM-L6-v2` (Galli et al., 2024) embeddings of completions and text query. For the effect of the re-ranking module ($w$ re-ranking in Table 2 (a)), we re-rank the initial image list using captions generated by an LMM. The LMM is prompted to focus on object actions and colors. Using the cloze generation and completion module, $w$ cloze completion attains +0.35% Recall@1 accuracy. Applying the comparison & re-ranking module, $w$ re-ranking achieves a +0.46% Recall@1 accuracy. Incorporating both modules, our full method achieves a +0.96% Recall@1 accuracy. These results show the effectiveness of the proposed components.

**Effect of the weighting strategies in comparing and re-ranking.** We conduct ablation studies on weighting strategies that assign re-ranking orders with corresponding scores (see Table 2 (b)). Two strategies are evaluated: exponential and linear decay. For exponential decay, $\beta$ ranges from 0.70 to 0.20, with $\beta =$0.50 achieving the best performance of 84.73% Recall@1 and 90.89% mAP accuracy. For linear decay, step sizes $d$ range from -0.30 to -0.10, with step size -0.20 performing best, achieving 84.73% Recall@1 and 90.88% mAP, slightly below the best exponential result.

**Effect of the choice of LLMs and LMMs in three steps of our method.** We conduct the ablation study on the choice of LLMs in different steps of our method (see Table 2 (c)). For cloze generation, we choose from Qwen3-8B and QWQ-Max. These two LLMs achieve similar performance, with QWQ-Max producing more reliable clozes for subsequent steps and achieving 84.73% Recall@1 and 90.89% mAP accuracy. For cloze completion, we choose among Qwen2.5-VL-7B, Qwen2.5-VL-3B, and QVQ-Max. We observe that LMMs with reasoning capability outperform those without it. Specifically, QVQ-Max achieves the best performance, while Qwen-VL-3B and Qwen-VL-7B deliver comparable results, *i.e.*, 84.22%, 84.13%, and 90.65%, 90.62% for Recall@1 and mAP, respectively. These findings show the importance of reasoning ability in cloze completion. For comparing and re-ranking, we choose among Qwen2.5-3B, Qwen2.5-7B, Qwen3-8B, and QWQ-Max. We find similar results that LLMs with reasoning capability have a better performance than LLMs without reasoning. QVQ-Max and Qwen3-8B achieve similarly strong performance, arriving at 84.52%, 84.73%, and 90.80%, 90.89% for Recall@1 and mAP, respectively. We consider that both LLMs effectively leverage the embedded world knowledge within the model to perform semantically equivalent comparisons among cloze completions. On the other hand, smaller-capacity LLMs without reasoning capability, such as Qwen2.5-3B, exhibit poorer performance of 83.67% Recall@1 and 90.37% mAP. We attribute this to their reliance on lexical-level comparisons. Due to the inherent diversity and ambiguity of natural language, these models tend to treat completions with similar meanings but different wording as distinct from the text query. Consequently, they often produce trivial or even incorrect ranking results.

**Effect of the candidate number of the initial ranking list.** We conduct an ablation study on the candidate number of the initial ranking list (see Table 2 (d)). Among the lengths of 2, 3, 4, and 5, re-ranking with a candidate list length of 3 achieves the best performance. Using lengths of 2 or 4 leads to a slight drop in performance, while increasing the length to 5 results in a further decline to 84.27% Recall@1, 90.66% mAP. These results suggest that a moderate initial list length of 3 provides a good balance between covering correctly paired candidates and reducing noise or ambiguity introduced by larger candidate sets.

**Effect of hyper-parameters in score balancing.** We present ablation results on the hyper-parameters for score balancing, namely $\alpha_1$ and $\alpha_2$ (see Table 2 (e), (f)). In Table 2 (e), with $\alpha_2$ fixed at 1, increasing $\alpha_1$ from 0.700 to 0.995 generally improves retrieval performance. Specifically, the mAP improves from 84.14% to 84.48% as $\alpha_1$ increases from 0.700 to 0.900, and continues to rise gradually, saturating at 84.66% when $\alpha_1$ is in the range of 0.910 to 0.995. In Table 2 (f), with $\alpha_1$ fixed at 1, varying $\alpha_2$ from 0.010 to 0.300 initially causes fluctuations in mAP, which peaks at 90.89% mAP when $\alpha_2 = 0.075$, followed by a decline below 89.00% mAP when $\alpha_2$ exceeds 0.100. These findings suggest that the first-stage score plays a crucial role in establishing a strong retrieval baseline, while the second-stage score contributes beneficial corrections within an appropriate range.

## 5 CONCLUSION

In this work, we present AnomalyLMM, the first framework to adapt LMMs for text-based person anomaly search. To address the challenges of fine-grained cross-modal alignment and anomaly recognition under sparse samples, our method introduces a novel coarse-to-fine pipeline and training-free adaptation strategies to bridge generative knowledge with discriminative retrieval. In particular, we are inspired by the cloze completion task to harness both LLM and LMM. We find that the correctly paired anomaly image provides crucial visual clues for LMMs. These cues guide LMMs to complete clozes that are semantically aligned with the input query. In contrast, when presented with unpaired visual candidates, LMMs lack the guidance from visual clues and are prone to producing dissimilar completions. Extensive experiments on the PAB benchmark validate significant improvements over competitive baselines, while qualitative analysis validates the interpretability of our anomaly-behavior alignment.

## 6 STATEMENT

### 6.1 ETHICS STATEMENT

We hereby solemnly declare that **we have carefully read the ICLR Code of Ethics, and that this research strictly follow these guidelines**. Our work primarily focuses on methodological advancements and is evaluated on standard, publicly available datasets (Yang et al., 2025). We have ensured that our use of these datasets complies with their respective licenses and terms of use. This research does not involve new data collection or direct interaction with human subjects. The datasets used have been previously anonymized by their curators, and our work does not introduce new risks to individual privacy or security.

### 6.2 REPRODUCIBILITY STATEMENT

To ensure the reproducibility of our research, we provide comprehensive details on our methodology, data, and experimental setup. All resources are intended to be self-contained and sufficient for reproducing the results presented in this paper.

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
