# OpenReview forum: "AnomalyLMM: Bridging Generative Knowledge and Discriminative Retrieval for Text-based Person Anomaly Search"
_ICLR.cc/2026/Conference — Submitted to ICLR 2026_

### Official Review · Reviewer_Gy4Y · 2025-10-29

**Soundness:** 3
**Presentation:** 3
**Contribution:** 2
**Rating:** 4
**Confidence:** 4

**Summary:**

This paper introduces AnomalyLMM, a training-free framework that leverages the advanced reasoning capabilities of existing LMMs for text-based person anomaly search. The authors propose a novel cloze-based re-ranking method, which encourages LMMs to fill in generated cloze-style text queries and rectifies the initial ranking list based on the filling performance.

**Strengths:**

1. **Good Writing**: The paper is well-written and the main contributions are clearly presented, making it easy for readers to follow the core ideas.
2. **Training-Free**: The proposed training-free post-processing method effectively leverages LMMs to refine the initial ranking list, which could greatly reduce the need for additional model training and resource consumption.

**Weaknesses:**

1. **Task Scope**: The specific problem that AnomalyLMM aims to solve within the text-based person anomaly search domain is not clearly articulated. The Introduction primarily emphasizes the challenge of discriminative retrieval, but such challenges are common across many tasks (e.g., text-based person retrieval). However, the experimental section lacks results on these related tasks. Furthermore, the proposed method appears to be well-suited for general vision-language cross-modal retrieval tasks, but the paper does not explore its generalization to these domains.
2. **Complexity**: The paper lacks a discussion on computational complexity. The proposed re-ranking method requires multiple LMM inferences for each text-image pair, which could be computationally expensive. According to the experimental results, when using X$^2$VLM as the baseline, the performance improvement is marginal. This raises concerns about whether the computational cost is justified.
3. **Experiments**: The difference between CMP and CMP-baseline is not clearly explained, and the experimental results show a large performance gap between them. Additionally, the paper lacks results for CMP+Ours, which would help clarify the effectiveness of the proposed method.
4. **Related Work**: The related work section lacks a discussion of existing research on using LMMs for retrieval tasks. This omission may affect the assessment on the novelty of the proposed approach.

**Questions:**

X$^2$VLM, as a generic vision-language pretrained model proposed in 2024, achieves the best performance after direct fine-tuning. What is the reason for this? Does this imply that the two main challenges described in the paper may not actually exist in this task (or in PAB dataset)? If so, the experimental results provided may not truly reflect the contribution to the text-based person anomaly search field.

**Details Of Ethics Concerns:**

No Ethics Concerns

---

### Official Review · Reviewer_2Qad · 2025-10-31

**Soundness:** 2
**Presentation:** 2
**Contribution:** 2
**Rating:** 4
**Confidence:** 3

**Summary:**

This paper proposes a training-free, coarse-to-fine framework that adapts Large Multimodal Models (LMMs) for text-based anomaly retrieval tasks. The core idea is to repurpose generative LMMs—traditionally designed for open-ended reasoning—into discriminative retrieval systems via a three-step cloze-based pipeline: (1) Cloze Generation masks key verbs and colors in textual queries using an LLM; (2) Cloze Completion prompts an LMM to fill these blanks based on image evidence; (3) Comparison & Re-ranking semantically compares completions with the original query to refine retrieval results. Experiments on the PAB dataset report modest improvements over strong baselines (e.g., +0.96% Recall@1) and claim enhanced interpretability. While the method is computationally efficient and avoids fine-tuning, its contribution is incremental: it mainly reuses existing LMM/LLM models with handcrafted prompt engineering and minor ranking fusion, offering limited novelty or empirical depth.

**Strengths:**

1.	Training-free design – The method does not require fine-tuning, making it efficient and easy to apply in low-resource or real-world settings.
2.	Coarse-to-fine pipeline – The hierarchical retrieval approach (initial ranking + re-ranking) improves interpretability and modularity.
3.	Novel use of cloze prompting – Reformulating retrieval as a cloze-completion task provides an intuitive way to guide LMM focus on fine-grained details.
4.	Empirical improvement on benchmark – Shows measurable, though small, gains on the PAB dataset, demonstrating some effectiveness.

**Weaknesses:**

1.	Limited novelty – The approach mainly combines existing LMM/LLM components and prompt engineering rather than introducing fundamentally new algorithms.
2.	Marginal performance gain – The reported +0.96% Recall@1 improvement is too small to justify the claimed contribution.
3.	Overly complex pipeline – The multi-step cloze generation, completion, and re-ranking process adds unnecessary complexity for a modest benefit.
4.	Weak experimental validation – Only one dataset (PAB) is used, with no cross-dataset or real-world evaluations to test generalization.
5.	No statistical significance analysis – Lacks error bars, confidence intervals, or repeated trials to confirm robustness of the reported results.
6.	Heavy reliance on proprietary models – Depends on specific LLM/LMMs (e.g., Qwen, QVQ-Max), making reproducibility and fairness questionable.
7.	Lack of ablation on design choices – The paper doesn’t analyze the contribution of each prompt design or masking strategy in depth.
8.	Limited scalability and efficiency discussion – The inference cost of multiple large model calls is not analyzed, contradicting the claim of deployment practicality.

**Questions:**

Please check weaknesses, and try to argue them, I will definitely read your response, good luck!

---

### Official Review · Reviewer_MucT · 2025-11-01

**Soundness:** 3
**Presentation:** 3
**Contribution:** 2
**Rating:** 4
**Confidence:** 5

**Summary:**

The paper introduces AnomalyLMM, a novel framework designed to address the challenges of text-based person anomaly search. The task involves identifying individuals exhibiting anomalous behaviors (e.g., falling, colliding) from video data using natural language queries. This task is difficult due to fine-grained visual-text alignment and the scarcity of anomaly samples in real-world datasets.

The authors propose a coarse-to-fine retrieval pipeline to adapt Large Multimodal Models (LMMs) for this discriminative retrieval task without requiring task-specific training or fine-tuning. The method leverages a three-step process:
Cloze Generation: This step converts the text query into a "fill-in-the-blank" prompt to isolate key verbs and color descriptors.
Cloze Completion: The model completes the masked query based on the visual content, focusing on relevant visual cues.
Comparison & Re-ranking: The LMM compares the query and image completions, ranking the candidates based on semantic alignment.
Experiments show that the method outperforms competitive baselines by achieving a +0.96% Recall@1 accuracy. Furthermore, the approach provides interpretability by offering clear alignments between textual queries and visual content, making it suitable for real-world deployment in public safety scenarios like smart surveillance.

**Strengths:**

No Fine-tuning Required: The proposed method is notable because it does not require additional fine-tuning, making it highly scalable and efficient for real-world applications, especially when anomaly data is scarce.

Coarse-to-Fine Retrieval: The use of a coarse-to-fine pipeline that combines both general retrieval models and fine-grained semantic reasoning through Cloze Generation and Cloze Completion significantly enhances the model’s ability to align nuanced textual queries with subtle visual anomalies.

**Weaknesses:**

Limited Evaluation with Diverse Datasets: While the paper presents results on the PAB dataset, a broader evaluation across multiple datasets would provide a clearer picture of the model's generalizability to various real-world scenarios. This is especially important since real-world anomaly data is often diverse and context-dependent.

Potential Hallucinations in Cloze Completion: While the Cloze Completion step is intended to help LMMs focus on relevant visual regions, there is still a risk of generating incorrect completions, especially when the image-query alignment is weak. The method constrains the model to output "UNKNOWN" when uncertain, but this may still limit its ability to handle highly ambiguous or novel anomaly behaviors effectively. A deeper analysis of failure cases could strengthen the discussion.

Dependency on Pre-trained Models: Although the method is training-free, it still relies heavily on pre-trained LMMs and text-to-image retrieval models. A more detailed exploration of potential biases or limitations introduced by these models (e.g., hallucinations, errors in alignment) would be beneficial.

Complexity of the Three-Step Process: While the approach is innovative, the three-step pipeline (Cloze Generation, Cloze Completion, and Comparison & Re-ranking) introduces complexity. A clearer breakdown of the computational costs and time complexity involved in each step would help assess its practical deployment feasibility.

**Questions:**

See Weaknesses

---

### Official Review · Reviewer_NhPv · 2025-11-26

**Soundness:** 3
**Presentation:** 3
**Contribution:** 3
**Rating:** 6
**Confidence:** 2

**Summary:**

This paper introduces AnomalyLMM, a training-free, coarse-to-fine framework designed to adapt LMMs for text-based person anomaly search. To address key challenges of fine-grained cross-modal alignment and sparse anomaly samples, the framework first uses a general text-to-image retrieval model to generate an initial candidate ranking. It then refines results through three core steps: Cloze Generation, Cloze Completion, and Comparison & Re-ranking.

**Strengths:**

1. The proposed AnomalyLMM adopts a training-free and coarse-to-fine retrieval pipeline, eliminating the need for costly fine-tuning of LMMs while improving zero-shot anomaly retrieval.

2. Through the designed three-step mechanism (Cloze Generation, Cloze Completion, and Comparison & Re-ranking), the framework can precisely perceive key semantic elements.

3. The proposed method provides highly interpretable visual-textual alignments without any task-specific training, surpassing the competitive baseline by +0.96%.

4. The writing of this paper is well, and the proposed method is easy to follow.

**Weaknesses:**

1. This paper only conducts experiments on PAB. Although it is a well dataset for text-based person anomaly search, more datasets can be introduced to validate the effectiveness of the proposed method. This raises concerns about its generalization to real-world scenarios beyond the PAB’s scenes.

2. Most compared methods are proposed before 2023. Some recently-proposed method in similar domain also can be used for comparison.

3. The ablation study results show that the number of initial candidates is strictly limited to 3. When increasing the size of it to 5, the performance of the proposed method will significantly drop in Recall@1 (from 84.73% to 84.27%). The proposed method may be sensitive to the selection of the number of Initial Candidate.

4. More cases can be shown in the supplementary materials.

**Questions:**

see weaknesses

---

### Meta-Review · Area_Chair_JxbE · 2025-12-26

**Summary:**

The reviewers unanimously appreciated the training-free design of the proposed method, and recognized that the clarity of the manuscript. However, they raised critical concerns with heavy dependence on pretrained LLMs--potential issues by hallucinations, reproducibility, and fairness  (MucT, 2Qad), high complexity (MucT, 2Qad, Gy4Y), lack of ablation study (2Qad, Gy4Y), marginal performance gain (2Qad), unclear application scope  (Gy4Y), and limited experiments (NhPv, MucT, 2Qad). The authors did not respond to these comments. The AC found that the concerns are valid and outweigh the strengths of the paper, and thus recommends rejection.

**Reviewer Concerns:**

The concerns below have not been addressed due to the lack of responses from the authors.

- Limited experiments, only on a single dataset (NhPv, MucT, 2Qad)
- Lack of comparisons with latest work (NhPv)
- Sensitivity to hyperparameters (NhPv)
- Heavy dependence on pretrained models (MucT, 2Qad, 2Qad)
- High complexity (MucT, 2Qad, Gy4Y)
- Marginal performance gain (2Qad)
- Lack of ablation study (2Qad, Gy4Y)
- Unclear application scope  (Gy4Y)

**Reviewer Scores:**

The reviewers were unanimously negative and the authors provided no response. Thus it is hard to expect any changes in reviewer scores.

---

### Decision · Program_Chairs · 2026-01-26

Reject